

# A comparative analysis on question classification task based on deep learning approaches

Muhammad Zulqarnain[1], Ahmed Khalaf Zager Alsaedi[2],
Rozaida Ghazali[1], Muhammad Ghulam Ghouse[1], Wareesa Sharif[3] and
Noor Aida Husaini[1]

[1] Faculty of Computer Science and Information Technology, Universiti Tun Hussein Onn
Malaysia (UTHM), Batu Pahat, Johor, Malaysia
[2] Physic Department, College of Science, University of Misan, Iraq
[3] Faculty of Computing, The Islamia University Bahawalpur, Bahawalpur, Punjab, Pakistan

## ABSTRACT

Question classification is one of the essential tasks for automatic question answering implementation in natural language processing (NLP). Recently, there have been several text-mining issues such as text classification, document categorization, web mining, sentiment analysis, and spam filtering that have been successfully achieved by deep learning approaches. In this study, we illustrated and investigated our work on certain deep learning approaches for question classification tasks in an extremely inflected Turkish language. In this study, we trained and tested the deep learning architectures on the questions dataset in Turkish. In addition to this, we used three main deep learning approaches (Gated Recurrent Unit (GRU), Long Short-Term Memory (LSTM), Convolutional Neural Networks (CNN)) and we also applied two different deep learning combinations of CNN-GRU and CNN-LSTM architectures. Furthermore, we applied the Word2vec technique with both skip-gram and CBOW methods for word embedding with various vector sizes on a large corpus composed of user questions. By comparing analysis, we conducted an experiment on deep learning architectures based on test and 10-cross fold validation accuracy. Experiment results were obtained to illustrate the effectiveness of various Word2vec techniques that have a considerable impact on the accuracy rate using different deep learning approaches. We attained an accuracy of 93.7% by using these techniques on the question dataset.

# INTRODUCTION

With the rapid development of computer technology and the internet, a huge amount of textual data in digital form are generated every day (*Wang & Qu, 2017*), and retrieve the given contents from a large amount of information rapidly and accurately. This has become an ordinary issue. Textual data is highly dimensional data having, comprising, consisting of irrelevant and unwanted features that are difficult to manage and maintain (*Sharif et al., 2017*). In natural language processing (NLP), the role of the question classification system is to predict the form of precise response according to the query.

Corresponding author
Muhammad Zulqarnain,
hi160052@siswa.uthm.edu.my

However, in most of the cases in the NLP, what the user desires to ask of the right answer to the question individually. One of the most appealing areas is the question answering (QA) system for both company and organization which provides more appropriate access to information than the conventional search engine. Question answering is one of the main technologies in a QA system that automatically seeks correct answers in natural language to random questions. For the information retrieval, the QA systems have been successfully applied in NLP and achieved significant results.

Furthermore, the implementation of a question classification system is identifying the category of sort questions that are asked in NLP. A QA system usually consists of four steps, semantic understanding, question classification, text retrieval, and answer extraction (*Le, Phan & Nguyen, 2015*). Most useful step is a question classification which can give useful information for subsequent execution and involves the customer category answer, the intent of question classification, and so on. The correlation between the questions and the category can be illustrated through a corresponding process:

$$F : X = \{C_1, C_2, \ldots, C_n\} \tag{1}$$

Where Turkish questions are represented by $X$, while $\{C_1, C_2, \ldots, C_n\}$ refers the set of categories, and $F$ defines the question, $X$ is classified into a specific category $C_i$ through some rules. In the significant part of the QA system, there are two key dialog implementations to determine the questions to the credible classification of the answers (*Mohd & Hashmy, 2018*). Identifying the questions is one of the defined matters according to the nature of the question. For instance, the question about the comparison is that "What is the difference between music and noisiness?" After categories, the questions of the QA system can perform the subsequent execution to improve the accuracy of the answer according to their intents. The other way of dialog implementation is to identify the questions based on the user requirement. For example, the question "Who is the chief of army staff of Malaysia?" is a question about a character or is a type of human (person). As per the question categorization, the QA system should usage the search technique specific to the human (person) type. Consequently, more efficient question identification can enhance the performance of the QA system.

Question classification is one of the correlated problems to documents categorization (*Ehsan & Mojgan, 2014*). However, in recent times, document categorization has been provided a huge quantity of scientific contribution. While question classification is mainly in the Turkish language now is a novel academic issue. The most important difference between question classification and document categorization is that the document dimension is much longer than the question dimension. Therefore, each character and word in question classification could be meaningful. As a results, generating features from a single question is more difficult than generating features from a large text (*Ehsan & Mojgan, 2014*). There are various approaches; machine learning, rule-based and hybrid approaches (*Razzaghnoori, Sajedi & Jazani, 2018*) have been used in the question classification tasks. In our research, some deep learning approaches such as LSTM, GRU, CNN, and their combination are utilized to classify user questions based on Word2vec techniques both Continue Bag of Words (CBOW) and skip gram. In this study, our main

contributions are: most articles associated to question classification concentrates on the English language and they have not studied an agglutinative language where the structure of words is generated by putting suffixes (morphemes) to the root of the word. The Turkish language has some distinctive features, which makes it problematic and has been demonstrated challenging for NLP. The majority of the challenges drives from Turkish complex morphology and how it deals with syntax.

Turkish, as a non-indo-European language, has several unique features, that make NLP difficult to determine. For example, the Turkish language does not have grammatical gender and noun classes (*Le, Phan & Nguyen, 2015*). In particular, it is very difficult to extract these nuances by NLP techniques, which have been generally applied for Indo European languages like English and German. Moreover, natural languages have several words that come from the same morphological class for structural and grammatical purposes. Particularly, the Turkish language, there are large number of word formations due to language structure. Turkish uses the derivative affixes and the inflectional to construct new words that typically produce a few hundred structures, and each verbal root can produce a million structures (*Ozturkmenoglu & Alpkocak, 2012*). For this reason, words that mean nearly a sentence in English language are more probable to be derived. When used contexts in a sentence, the Turkish words may provide a several derivational and inflectional suffixes for example;

dün + -di + biz + içinde + ofis →→ yesterday, we were in the office.
gor + onlar + görmek + yapabilmek + o →→ they were able to see it.
İngiltere + başbakanı + kim →→ who is the prime minister of England?

For this purpose, in most situations, the lemmatization process is very significant for obtaining the uninflected word forms in order to use IR (information retrieval) and other text processing techniques to tackle Turkish. The lemmatization method is commonly employed to enhance the efficiency of information retrieval systems. The main goal of lemmatization is to reduce inflectional forms and sometimes derivationally related forms of words to root forms. It establishes a connection between the surface form of connected words and dictionary forms. For above-mentioned reasons, in contrast to the English language, there is no useful lemmatization method in Turkish, due to the language structure of Turkish. This is another issue with the Turkish language when studying text processing (for more details about the difficulty of Turkish see *Oflazer (2014)*).

Another issue with Turkish is that it may neglect the tools required to determine the text information. However, we could not find any Turkish question datasets at the start of this study, so we decided to convert an English question dataset into a Turkish question dataset to see how the proposed approaches performed well in their best way.

On the other hand, *Mikolov et al. (2013)* introduced a new method of feature representation, which is applied in the feature extraction step is known as distributed representations of words or Word2vec (*Mikolov et al., 2013*). The concept behind the word representation approach is the terms with a semantic or syntactic connection which is used

with higher probability in a similar context (*Liu et al., 2017*). Therefore, the vectors of those words need a little bit close to each other if word1 and word2 contain similar contexts.

The learning algorithms of Word2vec are representing the words in a vector space and achieve superior results in the NLP mechanism to identify the relevant words. In the case of distributed representation of words, the neural network is very extraordinary to compute vector code of several linguistic regularities and patterns (*Mikolov et al., 2013*). In some cases, most models can be illustrated as linear representations instantaneously. For instance, the outcome of a vector computation vec("king")—vec("man") ? vec ("women") is closer to vec("queen") than to any other word vectors (*Mikolov et al., 2013*).

The main purpose of our research is a comparative analysis between the deep learning architecture and Word2vec method. Therefore, the major contributions of this study are presented as follows:

- Most articles focused on the English language have associated with question classification and they have not worked an agglutinative language where the construction of words is produced by assigning suffixes (morphemes,) to the root of the words. In NLP, the Turkish language has proved the preprocessing problem. As a non-Indo-European language, there are several unique features in Turkish languages that make NLP challenging.
- Another contribution is the impact of employing various Word2vec pre-trained word embeddings on various deep learning approaches. In our study, the first approach presented was to use Word2vec methods that are Continuous Bag of Words, skip gram to cluster words in the corpus and convert all words into vectors in the space. For the extraction of word vectors, the Word2vec method is applied to extract as a variation of the query word vectors of words. After that, the deep learning approaches such as CNN, GRU, LSTM, and their combinations including CNN-LSTM and CNN-GRU are applied for question classification. By using these four various approaches, the average correctness of CNN is 92.46%, LSTM achieved 90.89%, GRU obtained 91%, CNN-LSTM and CNN-GRU got 91.7% and 92.36% respectively over the Turkish question dataset.
- Moreover, there was no Turkish question labeled dataset as well, so in this study, we added a new Turkish question dataset which is translated from UIUC" English question dataset (*Li & Roth, 2002*).

## RELATED WORK

Question answering classification is an essential task of text classification. In the early 1950s, IBM was provided an environment to leading in examining text identification. Later, in the 1960s, Marun and Kahns introduced the keyword technique in the texts to automatically categorize the chosen texts.

There are three individual stages in the classic question answering system (*Ehrentraut et al., 2018*):

1. Question processing: this is an initial stage in questioning and answering systems where questions are asked by users (*Madabushi, Lee & Barnden, 2018*). The aim of this stage is

understood to apply the logical calculations for the representation and categorization of the questions.

2. Extraction and processing of documents: a set of relevant documents are selected in this stage and a set of paragraphs are captured which depend on the concentrations of the issue.

3. Answer processing: The purpose of this stage is considered to respond based on the relevant fragments of the documents. The preprocessing of the data requires pairing an answer based upon the similar contexts of the question asked. The general architecture of natural language for the question answering system is presented in Fig. 1 (*Athira, Sreeja & Reghuraj, 2013*).

Several different approaches have been successfully used in the question classification issue. Most of these approaches are divided into four groups: rule-based approaches, machine learning approaches, deep learning approaches, and hybrid approaches (*Razzaghnoori, Sajedi & Jazani, 2018*; *Hao, Xie & Xu, 2015*). Author Galitsky worked on rule-based approaches to classify questions based on the pair of questions with manually written rules according to the provided contents (*Galitsky, 2017*). While determining the particular rules is a massive time and struggle to process a variety of questions. However, the deep learning approaches, machine learning approaches, and rule-based techniques are capable to automatically construct a precise classification implementation utilizing different features of questions (*Ehsan & Mojgan, 2014*).

The machine learning techniques are superior to manual techniques discussed by authors (*Sarrouti & El Alaoui, 2017*). They described machine learning techniques give a reasonably easy way to classify questions as compared to manual techniques. Therefore, with this kind of implementation, the system can learn easily from the data and can be customized to a new system. On the other hand, there are limited studies that have used the hybrid approaches for the classification of the question. Here, we discuss a little bit about some research. The author (*Ehsan & Mojgan, 2014*) introduced a hybrid approach for a Persian closed area of question classification system, in which researchers willing a dataset that consists of 9,500 questions with the help of some researchers. They achieved a reasonable performance with an accuracy of 80.5% based on a large number of question classes.

Moreover, to resolve the question classification issues, there are several different machine learning approaches such as Neural Network, Random Forest, SVM, Decision Trees, Naive Base, KNN that have been applied for classifications. However, for the question classification task in NLP, the SVM ("Support Vector Machine") considers the key approach in machine learning (*Sherkat & Farhoodi, 2014*). To obtain their objectives, the authors *Sherkat & Farhoodi (2014)* used the SVM and dimension reduction method which uses a few linguistic features with a bag of the n-grams feature vector. Similarly, (*Huang et al., 2017*) applied a tree kernel with an SVM for identification the questions answering and successfully achieved an accuracy of 87.4% statistics. But during the experiment they did not use semantic and syntactic features. In the same way, the integrated approach is known as Hierarchical Directed Acyclic Graph (HDAG) with

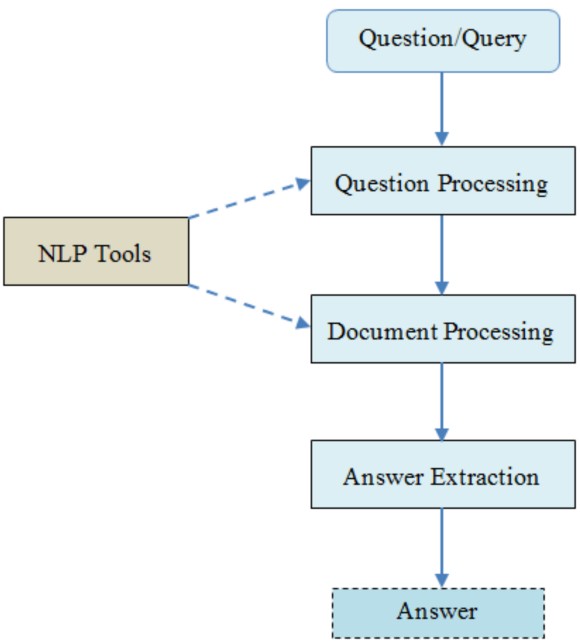

**Figure 1 The general architecture of NLQA system.**

kernel function implemented by *Chitra & Kalpana (2013)* the kernel function as known Hierarchical Directed Acyclic Graph (HDAG) which squarely perform some levels of chunks and their relatives on organized natural language database. Also, on the question answering corpus, Xu et al., was introduced a hierarchical technique based on the SNoW learning algorithm for the classification of the questions (*Xu et al., 2019*). In their research, they used a two-phase classification process. In the first phase, they presented the four most potential coarse-grained questions, classes. In the second phase, the question is classified into one of the child classes of the four coarse-grained question classes with an accuracy of 84.2%.

Furthermore, the author (*Merchant & Pande, 2018*) used the features roughly technique as a similar introduced by the authors of *Mohd & Hashmy (2018)*. Even though the enhancement they applied a dimensionality reduction method that is near to Principal Component Analysis (PCA) also known as Latent Semantic Analysis (LSA) to decrease the feature space for accurate classification. In their study, Back-Propagation Neural Networks (BPNN) and SVM are used. Their study presents that BPNN achieve superior results than SVM. On the other hand, to cluster words in the vocabulary (*Razzaghnoori, Sajedi & Jazani, 2018*) referred to some clustering algorithm in which perform to convert every question into a vector space. After that, Multi-Layered Perceptron (MLP) and SVM were used for the classification of the question. By evaluating the performance of such approaches, they obtain a reasonable accuracy of 73% using SVM and an accuracy of 72.52% by using MLP on 3 various datasets. They also prepared the UTQD-2016 dataset ("University of Tehran Question Dataset 2016"). In this corpus, many different types of questions taken from the jeopardy game are shown on official Iran's TV. In their third approach, they used the Word2vec method to convert each question into a matrix where

**Table 1 Summary of related researches.**

| Language | Dataset | Feature extraction algorithm | Classification approach | Accuracy (%) | References |
|---|---|---|---|---|---|
| English | UIUC question dataset | Bag of n-grams | SVM | 87.4 | (*Dell &Wee, 2003*) |
| English | UIUC questions dataset | Bag of n-grams | DT | 84.4 | (*Dell &Wee, 2003*) |
| English | UIUC question dataset | To produce more complicated features, such as named entities, Words, head chunks, chunks, POS tags, semantically related words and through these fundamental features few operators are utilized | SNoW | 91 | (*Close, 2002*) |
| English | NTCIR-QAC1 | Extract semantic information over named entities and words | SVM using HDAG kernel | 88.0 | (*Suzuki et al., 2003*) |
| English | UIUC questions dataset | WordNet semantic features for n-grams, word shape, headword, wh-words, headword | SVM and ME | 89 | (*Zhiheng, Marcus & Zengchang, 2008*) |
| English | UIUC | Headwords, word-shapes, associated words, hypernyms, Bi-grams, wh-words | NB | 93.8 | (*Loni, Seyedeh & Wiggers, 2011*) |
| Persian | UTQD.2016 | Word2vec, CBOW | LSTM, SVM, RNN and MLP | 81.77 | (*Razzaghnoori, Sajedi & Jazani, 2018*) |
| Persian | QURANIC questions | N-gram, POS tags, Lemma, Normalized word, special word detection and length of question | SVM with determined instruction | 80.5 | (*Ehsan & Mojgan, 2014*) |

every row presents a Word2vec representation of a word. After that, the authors used an LSTM (*Razzaghnoori, Sajedi & Jazani, 2018*) approach to classify the questions and reported 81.77% accuracy on three-question databases. The detailed summary of the related research in question classification is illustrated in Table 1.

# DEEP LEARNING APPROACHES

Deep learning approaches were derived from artificial neural networks and nowadays it is a principal area of machine learning and has successfully been applied to achieve excellent performance in various research areas. However, in the section, we evaluated four types of deep learning models for solving questions answering classification issues.

## Convolution neural network (CNN)

The Convolution neural networks are one of the most extensively used models in the deep learning community. And it is the same as a multi-layered perceptron, but it is effectively trained to combine sparsely attached convolutional layers with the completely connected dense layer. CNN consists of three major layers as convolution layer, Max-pooling layer, and fully connected layer for question classification are presented in Fig. 2. However, CNN utilizes only a single layer with variant functions to transfer information from one volume of activation to the next. The Convolutional layer is an essential layer of a CNN model that performs several computational processes. The max-pooling layer, in CNN, performs on data to compress and make it smooth. While for selecting the
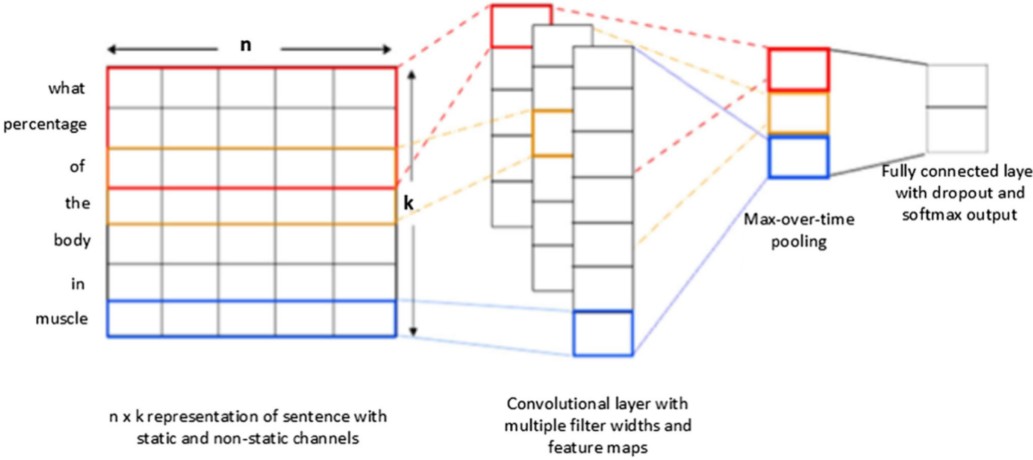

**Figure 2 Conventional CNNs architecture for question classification.**

maximum value of the responsive area the Max-layer is used which produces data-invariant small translational changes (*Zulqarnain, Ghazali & Hassim, 2019*). A fully connected layer is used as a final layer of CNN which produces the output by connecting all neurons in the forward and backward manner.

## Long short term memory (LSTM)

A Long Short Term Memory (LSTM) unit is a type of traditional RNN. It was initially introduced by German researchers Sepp Hochreiter and Juergen in 1997 (*Hochreiter & Schmidhuber, 1997*). LSTM approach is a variant of the traditional RNN that can learn long sequential data and maintain the propagation of error through all layers (*Samarawickrama & Fernando, 2018*). The LSTM contains special internal memory blocks and a gated mechanism that helps to solve the two popular drawbacks which are related to vanishing gradients or exploding in the conventional RNN. In LSTM the memory blocks consist of memory cells with self-connections and particular multiplicative units to handle the flow of information. An LSTM block consists of three gates including input gate, output gate, and forget gate (*Lee, 2015*). The architecture of the standard LSTM gates block is presented in Fig. 3.

In Fig. 3, the $(i_t)$, $(o_t)$, and $(f_t)$, are the input, output, and forget gates of LSTM through the time step t respectively, $(c_t)$ denoted the memory cell content, $?_t$ is the candidate state calculated in Eq. (5). $x_t$, $h_t$, and $h_{t-1}$ is the input, final output of the LSTM, and a previous time step of the hidden unit. Update the cell state vector is calculated as in Eq. (6). To perform the hidden state $(h_t)$ of an LSTM unit that is passed to the next sample in a sequence, the output of the output gate $o_t$, Eq. (3) is multiplied by the squashed cell state $c_t$ through tanh function in Eq. (7), where $W_{xo}$ and $U_{ho}$ are weight matrix, $b_o$ is a bias term, and Sigm $(x) = \frac{1}{1+e^{-x}}$.

$$i_t = Sigm(W_{xi}x_t + U_{hi}h_{t-1} + b_i) \tag{2}$$

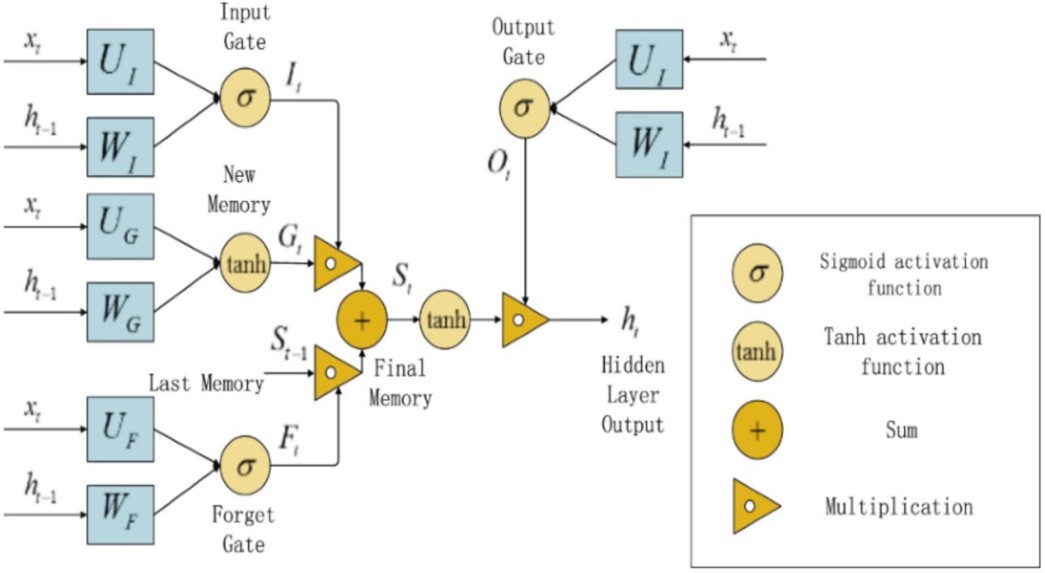

**Figure 3 LSTM Structure Diagram.**

$$o_t = Sigm(W_{xo}x_t + U_{ho}h_{t-1} + b_o) \qquad (3)$$
$$f_t = Sigm(W_{xf}x_t + U_{hf}h_{t-1} + b_f) \qquad (4)$$
$$\breve{a}_t = tanh(W_{x\breve{a}}x_t + U_{h\breve{a}}h_{t-1} + b_{\breve{a}}) \qquad (5)$$
$$c_t = f_t \times x_{t-1} + i_t \times a_t) \qquad (6)$$
$$h_t = O_t \times tanh(c_t) \qquad (7)$$

The weights and bias computed during the training process are $W_i, W_o, W_f, W \in R^{m \times p}$, $U_i, U_o, U_f, U \in R^{m \times m}$, $b_i, b_o, b_f, b \in R^{m \times 1}$. * is element-wise multiplication of two vectors. Here 'Sigm' is an element-wise logistic sigmoid activation function and 'tanh' is an element-wise hyperbolic tangent activation function.

## Gated recurrent unit (GRU)

The GRU is an advanced and simplified variant of LSTM that was initially proposed by *Cho et al. (2014)* on statistical machine translation. GRU is inspired by LSTM which controls the information flow inside the unit through update gate $z_t$ and reset gate $r_t$ without separate memory. Therefore, GRU has the capability of capturing the mapping relationship between time-series data (*Ghazali et al., 2014*; *Shen et al., 2018*) while it also has attractive advantages such as less complexity and efficient computational process. The architecture of GRU, which illustrates the relationship between update and reset gate is presented in Fig. 4.

However, GRU stores and filters information through internal memory capability and integrates the input gate and forget gate into a single update gate with the previous activation $\mathbf{h}_{t-1}$ and the candidate state represented by $\widetilde{\mathbf{h}}_t$. There are three major

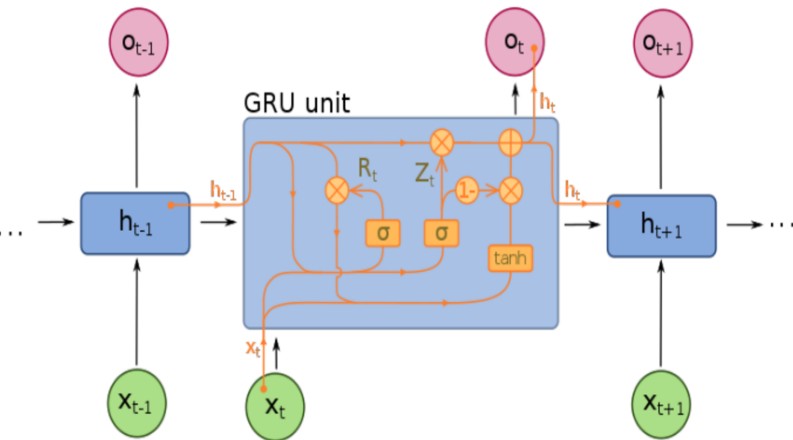

**Figure 4 GRU Architecture.**               

components of GRU are included update gate, reset gate, and candidate state and its equations are as follows:

$$z_t = \varphi(V_{xz}x_t + U_{hz}h_{t-1} + B_z) \tag{8}$$
$$r_t = \varphi(V_{xr}x_t + U_{hr}h_{t-1} + B_r) \tag{9}$$
$$\tilde{h}_t = tanh(V_{x\tilde{h}}x_t + U_{h\tilde{h}}(r_t \times h_{t-1}) + B_{\tilde{h}}) \tag{10}$$
$$h_t = (1 - z_t) \times h_{t-1} + z_t \times \tilde{h}_t) \tag{11}$$

Where $\mathbf{V}_{xz}$, $\mathbf{V}_{xr}$ and $\mathbf{V}_{x\tilde{h}}$ refer to the weight matrix among the input layer and update gate, reset gate, and candidate state while recurrent connection weight matrix is represented by $\mathbf{U}_{hz}$, $\mathbf{U}_{hr}$ and $\mathbf{U}_{h\tilde{h}}$ respectively. $\mathbf{x}_t$ is the time series sample input and hidden output is denoted by $\mathbf{h}_t$. $\varphi$ is the sigmoid activation function of update and reset gates, $^*$ performs element-wise multiplication operation and $\mathbf{B}_z$, $\mathbf{B}_r$ and $\mathbf{B}_{\tilde{h}}$ are the corresponding biases.

# PROPOSED METHODOLOGY

In the research methodology phase, the feature extraction methods have been briefly explained. These approaches are very important to identify the nature of the questions. After that, question classification approaches and classifiers would be examined. Besides, we illustrated modified deep learning architecture which has been utilized in this phase. In our proposed deep learning framework, we demonstrated the process of transforming words into vectors and identifying the questions to relevant classes. After that, for the question classification algorithm, we used Word2vec technique both skip-gram and Continues Bag of Words for classifying questions. In order to extract features, we used one mathematical expression of words. In this mathematical illustration, we allocated every word $x$ to a vector $f(x)$ such that if $x$ and $y$ have syntactic and semantic similarity then $f(y)$ and $f(x)$ will become nearby vectors.

In this study, we performed the multi-fusion CNN and RNN generated features to conduct the Turkish question classification. In the proposed methodology, we utilized two

various modified variants of the recurrent neural network models, such as LSTM and GRU with a combination of CNN.

## Modified LSTM

LSTM is different from standard RNN initially proposed by German researchers Sepp Hochreiter and Juergen in 1997 (*Hochreiter & Schmidhuber, 1997*) to learn long-term dependencies. We have explained the traditional LSTM architecture so far as shown in Fig. 3. But there is different LSTM architecture with various equations that help the creation of long-term dependency learning. All LSTMs are not the similar as the traditional ones. One popular variation of LSTM design that includes "peephole connections mechanism" is employed in our methodology (*Gers, Schraudolph & Schmidhuber, 2002*). In this section, we used a modified LSTM, which permits the gate layers to expression at the cell state and inserted the "peephole connection mechanism" which directly controls the gates defined as follows:

$$i_t = \varphi(W_{xi} \times [C_{t-1}, h_{t-1}, x_t] + b_i) \tag{12}$$
$$f_t = \varphi(W_{xf} \times [C_{t-1}, h_{t-1}, x_t] + b_f) \tag{13}$$
$$\bar{C}_t = tanh(W_{x\bar{c}} \times [h_{t-1}, x_t] + b_{\bar{c}}) \tag{14}$$
$$C_t = f_t \times C_{t-1} + i_t \times \bar{C}_t) \tag{15}$$

In these equations, where $x_t$ is the input transfer matrix of W, $C_{t-1}$ is the memory cell, component-wise multiplication is presented as x, while the hidden state vector $h_{t-1}$ and $\varphi$ shows the sigmoid function.

The output gate $o_t$ control the present hidden state value $h_t$, which uses memory cell content for the nonlinearity system result:

$$o_t = \varphi(W_{xo} \times [C_t, h_{t-1}, x_t] + b_o) \tag{16}$$
$$h_t = o_t \times tanh(C_t) \tag{17}$$

According to the following steps, the current stage of the hidden state $h_t$ is used for the acquisition of $h_{t+1}$. In other words, long short-term memory processes the word series recursively by computing their internal hidden state $h_t$ at each time step. The hidden activation of the final time step can be considered the linguistic description of the complete sequence and fed into the classification layer as input.

## Modified GRU

Initially, GRU was proposed by *Cho et al. (2014)* to extract the dependencies of the very recurrent unit at various time steps. GRU is a gating mechanism in RNN and is similar to LSTM with a forget gate. We have explained the traditional GRU architecture so far as shown in Fig. 4. But there is different GRU architecture with various equations that support the development of capturing long-term dependencies.

The mathematical expression of GRU is defined as follows:

$$z_t = \varphi(V_{xz}x_t + U_{hz}h_{t-1} + B_z) \tag{18}$$

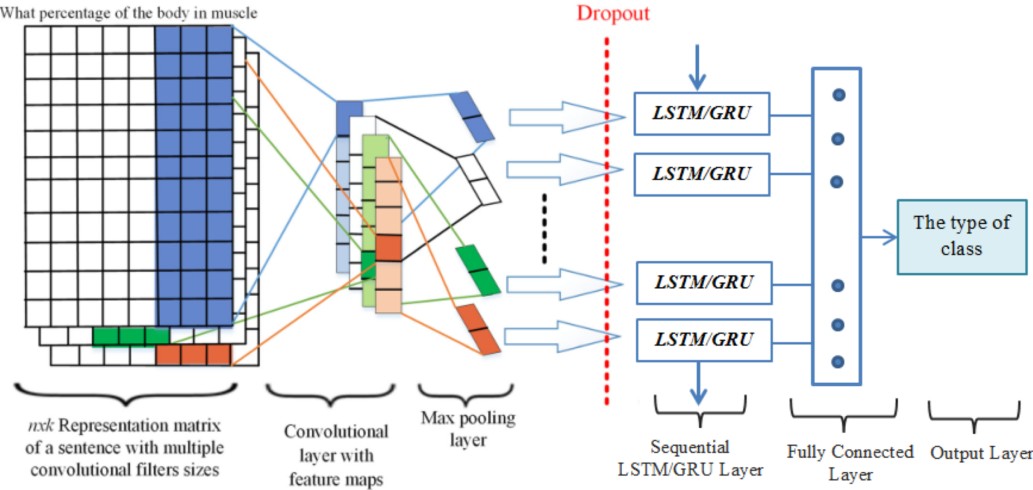

**Figure 5 CNN-LSTM/GRU architecture with an example.**

$$r_t = \varphi(V_{xr}x_t + U_{hr}h_{t-1} + B_r) \tag{19}$$
$$h_t = (1 - z_t) \odot h_{t-1} + z_t \tag{20}$$
$$\odot tanh(V_{x_t} + U(r_t \odot h_{t-1}) + B_h) \tag{21}$$

Where $h_t$ and $x_t$ are the output and input vector at time $t$. $z_t$ and $r_t$ are the update and reset gate vector. $\varphi$ and $\odot$ are the sigmoid activation function and element-wise multiplication operation while $B$ are the corresponding bias.

## CNN-LSTM/GRU models

In this section, we proposed a deep learning hybrid architecture which includes the following parts: word embedding with the Word2vec methods, convolutional neural network, and recurrent neural network with its variants, such as LSTM and GRU, while fully connected layer employed as a softmax output. The word embedding method is applied to convert input text into numerical word vectors to translate into CNN and RNN models. In this way, several convolution kernels of various dimensions were used to capture more helpful features in question classification. With this technique, CNN preserves the temporal data and generates a single value using the max-pooling layer. Similarly, the RNN layer is utilized to obtain the temporal features at the input level and captures long-term dependencies. In word embedding, every question was illustrated as a word embedding matrix to create a classifier using a CNN. Provided a question consisting $n$ words $v_1, v_2, v_3, \ldots v_n$ each word with its pre-trained $d$-dimension word embedding matrix is swapped, and stacked row-wise to produce occurrence matrices $V_i \in R^{n \times d}$. Moreover, in this study, we trained the hyperparameters using $ReLU$ in the deep learning approaches. Specifically, we selected $ReLU$ because ReLU ("Rectified Linear Unit") ReLU has been widely used in advanced deep learning architecture like CNN, RNN. Fig. 5, shown the essential architecture of the proposed approaches which is

adapted from (*Yang, Macdonald & Ounis, 2018*; *Alayba et al., 2018*), and it summarizes the combination of three various deep learning approaches such as CNN, LSTM and GRU. A quick overview of each layer will be described in more detail.

### Input layer

The input layer is selected as an initial point of the networks. Let describe the total number of unique words with $w_1$, $w_2$, $w_3$,…, $w_n$ in the dictionary $D = d_1, d_2, d_3, …. d_m$. To identify the questions, the Word2vec model is used to translate any word in the query into a particular vector with one of the fixed sizes of 100, 200, 300 or 400. Every question $j$ is presented with a two-dimensional $n \times k$ matrix $c_j = [v_1, v_2, …, v_n]$, the description of $k$ refers to the dimensionality of the $vi$ embedding and a large number of words is denoted by $n$. To provide the same length for all questions the input layer transmits data samples as a sequence of unique indices of similar dimensions.

## CNN LAYERS

The convolution layer is the most useful and basic layer of CNNs that perform the convolution process in the form of row representation through word vectors obtained from the embedding layer. CNN layers contain a set of learnable filters or kernels which map to produce two-dimensional activation. Let's considered the $h$ words at time $t$ with weight matrices w of dimension $w \in R^{h \times m}$ to perform the following convolutional computation:

$$c_i = f(X_{i+h-1} \times w + bi) \tag{22}$$

Where $f$ refers to the non-linear *Relu* activation function and the feature map generated to represented by $c_i \in R_{n-h+1}$ with $h$ words every time frequently, while bias term denoted by $b_i$. After that, the max-pooling layer performs to receive created features from convolution which change the features map into its maximum activation value, as follow:

$$P_i = maxC_i \tag{23}$$

where, $Pi \varepsilon R^{n-h+\frac{1}{2}}$ refers to the new feature map in order to obtain the various levels of features from the convolutional layer, we selected three convolutional layers with filter windows of different sizes 3, 4, and 5. After performing the max-pooling operations, we combined the features from various levels of convolution layers to achieve the final multilevel feature combination output as presented in Fig. 5.

### Feature mapping of CNNs layer with RNNs layer

Each input has a vector series, which scans it with a fixed filter distance. In this technique, the filter sizes of 3, 4, and 5 are used to carry the features of words. The CNN's layers efficiently reduce the input features vector, and give the better-compressed presentation through the max-pooling layer as compared to original raw features and the output generated by CNN's layers are further processed as inputs to the RNNs layers passing through the gating mechanism for learning the high informative features. Furthermore, in

order to classify questions every layer processes different features in a question with the *Relu* activation function in the feature map.

### RNN layers

RNN layers give exhibit temporal dynamic behavior (*Choi et al., 2017*) which processes the sequential data within the network. The recurrent layer has the capability to capture the long-term dependencies; therefore we feed original word embedding as an input to the RNN layer instead of those features generated by CNN. The purpose of selecting the RNN layers approach is to take the sequence data through utilizing the previous information. In these RNN layers, the final output of the layers has the equivalent number of units.

Due to sequence data, the RNN layers can learn temporal features from it. After this, we performed a different combination of CNN and RNN acquired features to carry out question classification. Based on this technique, sequential features should be perfectly maintained and a sequence created by max-pool layer instead of a single value. According to the following process, the data are fed into an RNN layer with many to one mechanism and a fully connected layer with softmax output.

## EXPERIMENTAL DESIGN & DATASET

We conducted an experimental design to evaluate the effectiveness of five different deep learning approaches on the Turkish question database. Therefore, this section briefly describes the Turkish question database and experimental settings. The overall architecture of this research is presented in Fig. 6.

### Question database description

We evaluate the performance of proposed deep learning approaches on the Turkish dataset for question classification. There is an absence of a Turkish question database as compared to the English database. In this research, we used a dataset of Turkish questions that is adapted from an English Question Dataset that has been used by *Li & Roth's (2002)*.

They referred to two-layered classification, which is extensively applied for question categorization. This dataset includes six offensive classes and fifty fine-grained classes that are reported as 'offensive fine' including "LOCATION: city". This dataset is divided into two parts are training and testing. We experimented to use 5,400 questions for training data and the remaining 600 questions for testing data. The distribution of this dataset (*Le, Phan & Nguyen, 2015*) categorized into main classes and sub-classes are reported in Table 2. In our experiments, we reconstruct the Turkish dataset from the English dataset.

### Experimental setting and hyperparameters

Deep learning based approaches have the ability to acquire complex relationships among inputs and outputs (*Srivastava et al., 2014*). In our experiment, we applied Adam optimizer to set their default optimal parameters setting with a learning rate of 0.005 and decay factor is 0.9. For the CNN layers, we applied three channels where each one uses a two-dimensional convolutional layer with kernel window size 3, 4, and 5. We used the rectified linear unit (ReLU) activation function for each convolutional layer. For each iteration of the training procedure, we fix the batch size to 32. For a fair comparative

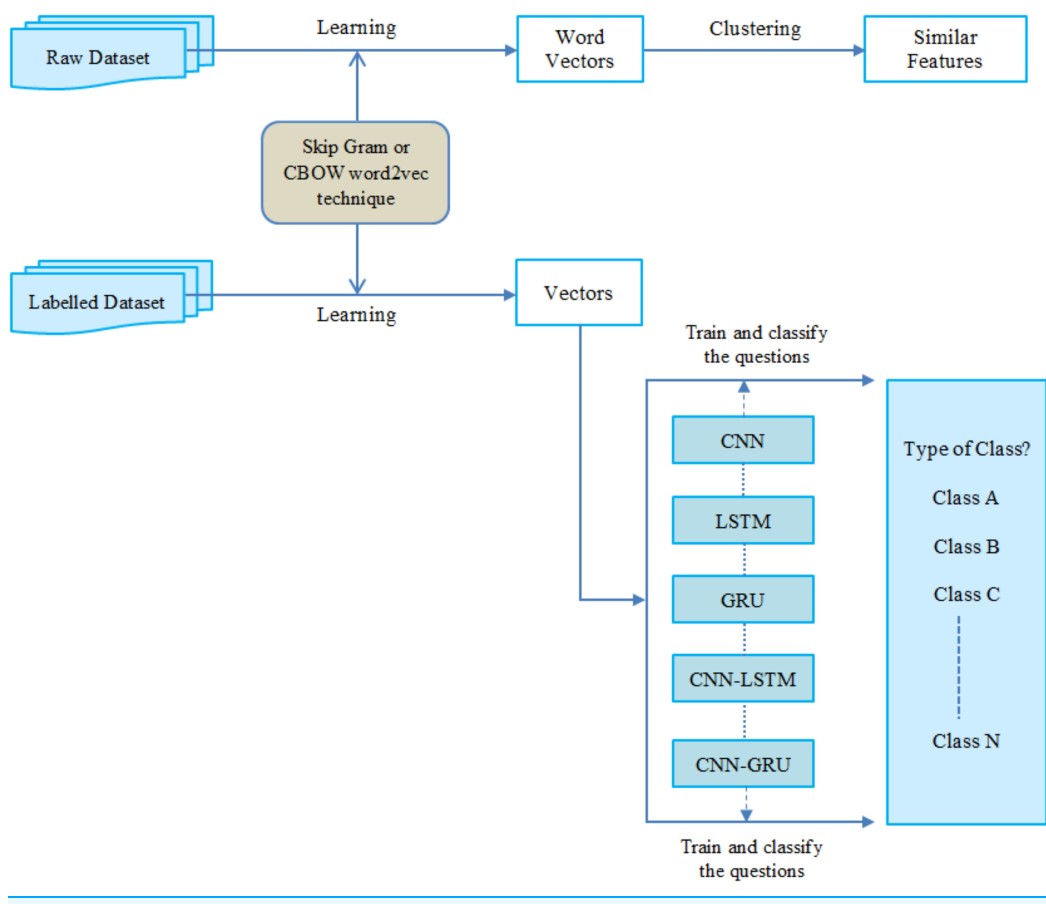

**Figure 6  The general architecture of this study.**

analysis, a few preprocessing steps are performed to improve the quality of the dataset. However, during the training process, many connections are involved as a result of sampling noise; while it did not exist in the test data. This problem may conduct to overfitting and minimize the prediction ability of the network (*Srivastava et al., 2014*). For this issue, we applied the dropout method to reduce the overfitting with the dropout probability of 0.2 for recurrent layers and 0.5 after the convolution layer. Moreover, for training the proposed models, we used Cross Entropy with $L_2$ regularization as minimizing the loss function, which referred to as follows:

$$J(w, b) = -\frac{1}{2}\sum_{i=1}^{m}[y_i \log \hat{y}_i + (1 - y_i)\log(1 - \hat{y}_i)] + \lambda 2m \sum_{l=1}^{m}||w||_F^2 \tag{24}$$

Where $y_i$ is refer ground truth and classification probability for each class represented by $\hat{y}_i$. We set $w = 0.001$, of Frobenius norm value by compressing $L_2$, which is the coefficient for $L_2$. During the training process, the result presents the $L_2$ regularization and dropout method can perform better to avoid overfitting. Table 3, provides the optimal values of hyperparameters, which have been applied for the training of the proposed framework.

**Table 2 Distribution summary of question categories in the UIUC Question dataset.**

| Category | Training | Testing | Category | Training | Testing |
|---|---|---|---|---|---|
| DESCRIPTION | 1159 | 96 | HUMAN | 1,218 | 68 |
| Reason | 190 | 8 | Other | 216 | 12 |
| Manner | 274 | 2 | Description | 26 | 4 |
| Definition | 422 | 124 | Group | 48 | 8 |
| ABBREVIATION | 84 | 9 | Creative | 208 | 0 |
| Currency | 4 | 4 | Animal | 110 | 15 |
| Religion | 4 | 0 | Individual | 190 | 58 |
| Plant | 14 | 6 | Title | 960 | 1 |
| ENTITY | 1252 | 96 | LOCATION | 836 | 82 |
| Body | 15 | 3 | City | 130 | 20 |
| Letter | 9 | 0 | State | 70 | 10 |
| Instrument | 10 | 2 | Mountain | 24 | 5 |
| Symbol | 11 | 3 | Country | 152 | 4 |
| Lang | 16 | 2 | NUMERIC | 900 | 226 |
| Technique | 34 | 6 | Temp | 8 | 4 |
| Word | 25 | 3 | Date | 221 | 48 |
| Vehicle | 28 | 5 | Weight | 12 | 4 |
| Product | 40 | 5 | Code | 8 | 2 |
| Substance | 40 | 14 | Speed | 10 | 5 |
| Color | 40 | 10 | Period | 30 | 9 |
| Food | 102 | 4 | Size | 16 | 0 |
| Term | 94 | 8 | Distance | 35 | 15 |
| Sport | 62 | 1 | Money | 70 | 4 |
| Event | 58 | 4 | Count | 368 | 9 |

## Word2vec models

The Word2vec comprehends and vectorizes the meaning of a word in a document established on the hypothesis that words with comparable meaning in a provided context display near distances (*Sahlgren, 2008*). It is an open-source platform offered by Google in 2013 under the Appache License 2.0. Fig. 7, demonstrates the model architectures of CBOW and Skip-gram learning procedures of Word2vec was referred to by *Neelakantan et al. (2015)* and *Yang et al. (2017)*. Input, Projection and output layers are present in all the learning algorithm, while their output processes are different. $W_n = \{W_{(t+2)}, W_{(t+1)}, \ldots, W_{(t+1)}, W_{(t+2)}\}$ is received as arguments for the input layer, where $W_n$ shows words. The projection layer is a multidimensional vector array that stores the sum of various vectors. The output layer matches the layer which outputs the results of the vectors from the projection layer. It is a shallow of two-layer neural networks that are educated to perform word embedding method. Specially, CBOW is similar to the feedforward Neural Network Language Model (NNLM) (*Armeni, Willems & Frank, 2017*) and predicts the output word from other near word vectors. The algorithm of the Word2vec model extracts features from a provided

**Table 3 Summary of major parameters utilized in the deep learning approaches with Word2vec models.**

| Parameters | Value | Parameters | Value |
|---|---|---|---|
| Dimensionality | 100, 200, 300, 400 | Dropout | 0.2, 0.5 |
| Sample | 0.001 | Seed | 1 |
| Window size | 10 | min_alpha | 0.0001 |
| Sentences | None | min_count | 5 |
| Batch_words | 10,000 | cbow_mean | 1 |
| Max_vocab_size | None | Alpha | 0.025 |
| SG[0,1] | 1 for skim gram otherwise CBOW | null_word | 0 |
| HS[0, 1] | 0 | Activation function | ReLU |
| Negative | 5 | trim_rule | None |
| Hashfxn | 5 | sorted_vocab | 1 |

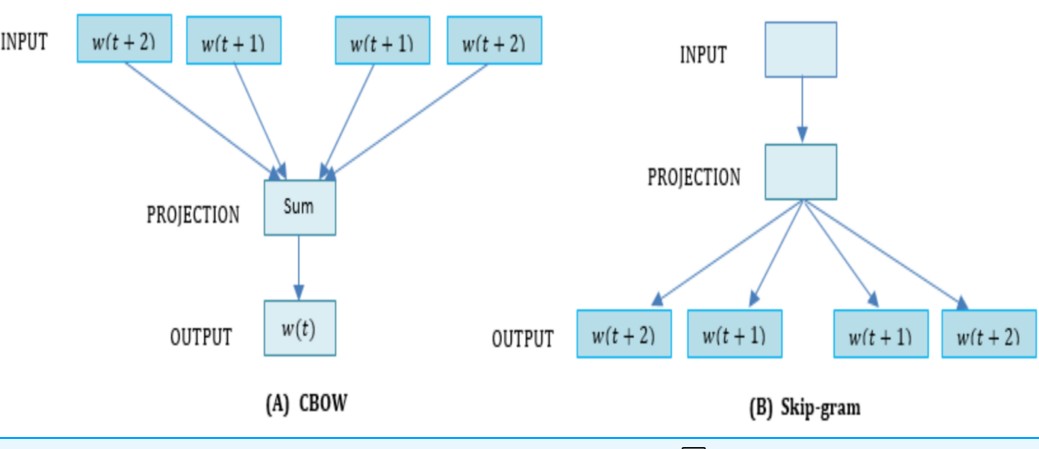

**Figure 7 The (A) CBOW and (B) Skip gram model.**

text corpus without any intervention from a human expert. Most essentially, if the text size is too small or only a separate word, it performs quite well. By providing a big corpus, it generates word vectors from a large number of texts and makes it appropriate by comparing the contextual data the input words performed similarly as shown in (https://github.com/akoksal/turkish-word2Vec). In the Word2vec space, every special word in the text is allocated to a connected vector (https://israelg99.github.io/2017-03-23-Word2Vec-Explained/). The meaning of words is one of the most significant intents in deep learning that are entirely accomplished with employing the Word2vec for classifying major entities (*Mikolov et al., 2013*). For learning word embeddings from raw data, it is a computationally well-ordered predictive framework. There are two different techniques of Word2vec, as follows:

1. Skip gram
2. CBOW (Continuous Bag of Words)

Algorithmically, these are two approaches near to each other *Buyukkinaci (2018)*. By computationally, Continuous Bag of Words (CBOW) is a continuously distributed word representation approach, which classifies the core words (target) based on the neighboring words. The fundamental principle of CBOW includes identifying when a given word comes from neighboring words analysis. The CBOW architecture shows the advantage that the information in the dataset is organized uniformly. Furthermore, the CBOW method derives the dictionary $\|V\| \in R^m$ by mapping the words $(c_1, c_2, \ldots, c_t)$ in the corpus to the projection layer. Then, in the projection layer the corpus word $c_t$ is mapped to the unique position $w_t$ and refers to the context size byk. It is a methodological result that the corpus terms are read sequentially by Continuous Bag of Words $[c_{t-k}, c_{t-k+1}, \ldots, c_{t+k}]$ and achieves the corresponding word position $[w_{t-k}, w_{t-k+1}, \ldots, c_{t+k}]$ in the projection layer by a hash table. Finally, it performs the following operations on the Context $(w_t)$ of $w_t$, where $V_t$ is the context accumulated sum of $w_t$ mentioned in Eq. (14). As one observation, CBOW processes entire contexts sequentially.

$$V_t = \sum_{t-N}^{t+N} Context(w_t) \tag{25}$$

On the other hand, the skip-gram algorithm is very similar to CBOW as presented in Fig. 7. The major difference between them is which swaps the output and input, and use the reverse functionality to each other. Skip-gram is typically used to predict all background terms in a single input word for a given target word. The projection layer of the Skip-gram predicts next words around the inserted into the input layer. The Skip-gram architecture shows the advantage of vectorizing when new words appear. Generally, the training purpose of the skip-gram method usually discovers word vectors that help to identify near words in the relevant contexts (*Nooralahzadeh, Øvrelid & Lønning, 2018*). The skip-gram model shows a mechanism to predict a one-word vector from other words ("opposite of Continuous Bag of Words"). We summarize it and describe our notations for the execution skip-gram approach. Suppose words of corpus and their contexts: $D = (w, c) = (w_1, c_1), (w_2, c_2) \ldots (w_n, c_n); w_i \in V_W, c_i \in V_C$, which are one-hot vectors, where $V_W$ is present the word vocabulary and $V_C$ is present the context vocabulary with sizes of $S_W$ and $S_C$ respectively. The learning procedure of skip-gram (SG) attempts to train the contextual distribution for separate word by optimizing the prospect function as follows:

$$SG(\mathbf{c}/\mathbf{w}; E, F) = \prod_{i=1}^{n} SG(c_i/w_i; E, F) = \prod_{i=1}^{n} \frac{exp(w_i^T EF)c_i}{\sum_{\bar{c} \in C} exp(w_i^T EF)\bar{c}} \tag{26}$$

Where $E$ and $F$ are the parameter matrix of the shapes $(S_W \times d)$ and $(d \times S_C)$ respectively. $d$ is the dimensionality of the embedding vector space. Contrary, $P(x_{io} = 1 \mid w_i, c_i; E, F)$ presents specific contexts that appear close to a word or not. While, in this study, we refer $P(D; \theta)$ for the skip-gram training demonstrated by $P_{SG}(\mathbf{c} \mid \mathbf{w}; E, F)$. The skip-gram algorithm works more effectively on a big corpus, every context-center pair is handled as

a new consideration. The architecture of skip-gram (sg) and Continuous Bag of Words (CBOW) are shown in Fig. 7.

## Training Word2vec embedding model

As part of this research, we have the selected most widely applied deep learning architecture with the Word2vec model that focuses on question classification. For word embedding, we implemented various parameters in order to train Word2vec of both Skip gram and CBOW models on Wikipedia corpora. We explored and trained our Word2vec models on a Turkish Wikipedia dataset for question classification. As the largest encyclopedia in which documents are well organized by topics on the Internet, we preferred it as the dataset. Therefore, Wikipedia corpora are well suitable for the analysis of the Word2vec approach. However, in our study, we eliminate unnecessary words from the question classification corpus with less than 5 words during the training on Wikipedia corpus. Because these words have minor quantities of data which are usually helpless for training the Word2vec model (e.g., some have stop words, streaming, and emotions). Moreover, we conducted an experiment by using Wikipedia corpus, to build our skip-gram and CBOW model with various vectors lengths 100, 200, 300, and 400. In addition, to avoid the overfitting issues, we used the dropout technique (*Hinton, 2014*), with a dropout rate of 0.5 for convolutional layers while 0.2 for recurrent layers.

For all the experiments, to train our Word2vec model, we used Gensim (*Liu et al., 2018*) to generate a set of word embeddings by setting the context window size *W*, the dimensionality *D*, the complete number of negative samples is presented by *ns* and the skip-gram is shown by *sg*. The value of the predefined parameter for the context window size is selected as W = {5}. Similarly, for this window size, to explore both of the high and low dimensions, we applied four various dimensionality sizes *D* = {100, 200, 300, 400} for the word2vec vectors. Therefore, we have chosen the default parameters for the training of deep learning approaches with the word2vec model. We fixed 5 as the negative sampling, batch words to 10,000 minimum counts of words to 5 and iteration to 5. In addition, we analyzed the effect of the dimensional vectors on the Turkish question dataset, while it includes near about one million Turkish articles and has been explored on Turkish Wikipedia. After removing the noisy words that are frequently less than 5, more than 200 thousand Turkish words are collected in the database. The parameters of deep learning approaches employing the Word2vec models are presented in Table 3.

Additionally, in this study accuracy was selected as an evaluation metric. It evaluates the correctness of the model and is calculated as the ratio of correctly classified instances (TP) divided by the total numbers of instances (TP + FP) on the entire dataset. The formula of accuracy in question classification defined as follows:

$$\text{Accuracy} = \frac{TP_{(n)}^{(m)} + TN_{(n)}^{(m)}}{TP_{(n)}^{(m)} + FP_{(n)}^{(m)} + FN_{(n)}^{(m)} + TN_{(n)}^{(m)}} \tag{27}$$

Where, TP and *TN* present True Positive, True Negative respectively, which indicate the correct classification for the relevant class while *FP* and *FN* refer to the False

Positive, and False Negative which determine the false classification for the relevant class. Particularly, in word embedding, we utilized the Word2vec model both of skip-gram and CBOW as feature extraction models. Furthermore, in terms of 10-cross validation accuracy, we also compared the results.

## EXPERIMENTAL RESULTS

In this part, we illustrate the performance of different deep learning algorithms using Word2vec embedding vectors of both CBOW and the skip-gram methods on the question dataset. The semantic and syntactic connections between words can be efficiently captured by these techniques. In this way, initially, the Word2vec model calculates word vectors in the vocabulary words. Similarly, the word2vec model initializes and selects random vectors from word vectors. Then, these algorithms attempt to increase the cosine similarity between all terms and their contexts, which is described based on the method. Consequently, these algorithms will be capable to allowing word vectors from a large amount of text of the Wikipedia corpus, while their nearness is associated with their corresponding words.

These deep learning techniques applied for questions classification is based upon the Word2vec models of skip-gram and CBOW with random vector approaches. To the best of our knowledge, this research concentrates on agglutinative language even this is the first time studied in an agglutinative language in detail for question classification. As a performance analysis, we experimented based on the word2vec models and achieved satisfactory results in a term of accuracy. Tables 4 to 8 compare the results of deep learning models with both word2vec word embedding techniques as a function of training volume based on the fixed epoch in question classification.

## DISCUSSION

This section analyzes the competitiveness and effectiveness of our proposed results with various deep learning approaches with both word2vec word embedding techniques by taking into consideration an accuracy. Tables 4 to 8, show the accuracy comparison of deep learning models based on the various number of feature vectors with Word2Vec methods. The previous research on question classification focuses on various tasks of occurrence for relevant class categorization or named entities (Derici et al., 2015). However, they integrated a rule-based method employing an HMM-based sequential categorization method (Donmez & Adalı, 2017). And their answers could not generalise to question classification tasks in an agglutinative language. Furthermore, some alternate studies have been performed on similar language work for questions classification (Razzaghnoori, Sajedi & Jazani, 2018); (https://github.com/thtrieu/qclass_dl/blob/master/ProjectDescription.pdf, https://github.com/thtrieu/qclass_dl/blob/master/ProjectPresentation.pdf) has not examined the influence of the Word2vec models on both variants skip-gram and CBOW. Particularly, to evaluate the performance of question classification, they applied some parameters such as feature vector size, window size. In this study, the experimental results demonstrate the factors that those mentioned-above can certainly affect the performance of the question classification system.

**Table 4** The results of CNN model using Word2vec.

| Number of feature vectors | Type of Word2vec model | Accuracy (Test) | Accuracy (10-cross fold validation) |
| --- | --- | --- | --- |
| 100 | CBOW | 92.1 | 86.0 |
| 100 | Skip gram | 92.4 | 88.8 |
| 200 | CBOW | 92.2 | 86.3 |
| 200 | Skip gram | 92.1 | 88.9 |
| 300 | CBOW | 91.9 | 87.6 |
| 300 | Skip gram | 93.7 | 90.1 |
| 400 | CBOW | 91.8 | 87.0 |
| 400 | Skip gram | 93.5 | 89.2 |

**Table 5** The Results of LSTM using Word2vec model.

| Number of feature vectors | Type of Word2vec model | Accuracy (test) | Accuracy (10-cross fold validation) |
| --- | --- | --- | --- |
| 100 | CBOW | 90.9 | 86.9 |
| 100 | Skip gram | 91.2 | 87.0 |
| 200 | CBOW | 90.8 | 87.9 |
| 200 | Skip gram | 90.7 | 87.5 |
| 300 | CBOW | 90.4 | 87.1 |
| 300 | Skip gram | 91.3 | 88.7 |
| 400 | CBOW | 91.0 | 88.6 |
| 400 | Skip gram | 90.8 | 88.2 |

**Table 6** The Results of GRU using Word2vec model.

| Number of feature vectors | Type of Word2vec model | Accuracy (test) | Accuracy (10-cross fold validation) |
| --- | --- | --- | --- |
| 100 | CBOW | 91.5 | 86.5 |
| 100 | Skip gram | 91.2 | 89.3 |
| 200 | CBOW | 90.7 | 86.7 |
| 200 | Skip gram | 90.3 | 89.5 |
| 300 | CBOW | 90.6 | 88.6 |
| 300 | Skip gram | 92.0 | 87.9 |
| 400 | CBOW | 91.5 | 88.7 |
| 400 | Skip gram | 90.2 | 88.6 |

In our research, generally, we examined four different deep learning models including CNN, GRU, LSTM, CNN-LSTM, and CNN-GRU based on Word2vec models of both CBOW and skip gram. However, by comparison, the CNN, CNN-LSTM and CNN-GRU models are capable to achieved significantly superior results in the term of accuracy when using skip-gram model on Turkish questions classification dataset as compared to CBOW model (Tables 4, 7 and 8). On the other hand, CNN, CNN-LSTM, and CNN-GRU, commonly perform better than LSTM and GRU architectures by using CBOW model.

**Table 7 The Results of CNN-LSTM architecture.**

| Number of feature vectors | Type of Word2vec model | Accuracy (test) | Accuracy (10-cross fold validation) |
| --- | --- | --- | --- |
| 100 | CBOW | 91.1 | 86.4 |
| 100 | Skip gram | 90.9 | 87.9 |
| 200 | CBOW | 91.8 | 88.8 |
| 200 | Skip gram | 93.0 | 89.1 |
| 300 | CBOW | 90.7 | 87.0 |
| 300 | Skip gram | 92.2 | 88.7 |
| 400 | CBOW | 91.5 | 87.3 |
| 400 | Skip gram | 92.4 | 89.2 |

**Table 8 The Results of the CNN-GRU model.**

| Number of feature vectors | Type of Word2vec model | Accuracy (test) | Accuracy (10-cross fold validation) |
| --- | --- | --- | --- |
| 100 | CBOW | 91.7 | 84.7 |
| 100 | Skip gram | 92.8 | 88.3 |
| 200 | CBOW | 93.2 | 86.7 |
| 200 | Skip gram | 92.9 | 89.5 |
| 300 | CBOW | 91.4 | 87.2 |
| 300 | Skip gram | 92.5 | 89.6 |
| 400 | CBOW | 91.8 | 86.7 |
| 400 | Skip gram | 92.6 | 89.5 |

In most of the cases, the CNN-LSTM and CNN-GRU approach achieved better results based on skip-gram than CBOW. Moreover, we have observed an excellent result in CNN approach, an accuracy of 93.7%, based on the skip-gram with 300 feature vectors. Furthermore, we experienced that when utilizing the correct form of a dataset can probably incorporate more vocabulary for the question classification database. For this reason, the correlation between corpus and the classification dataset provides better question-level representation. Finally, on the same dataset, we compared the performance of our proposed approaches with a similar study (https://github.com/thtrieu/qclass_dl/blob/master/ProjectDescription.pdf), in which the researchers have used the LSTM approach to obtain 94.4% accuracy in English language; we noticed that our achieved results were low compared to this study carried out in English. The main cause behind of this is the Turkish language construction as we already described above in the introduction section. As a result, there is a lack of efficient Turkish Language lemmatization tools compared to English language.

## CONCLUSIONS

Question classification is an important area in NLP. Recently, there are several deep learning models have been used to solve these issues and have shown remarkable results in NLP. In this study, we applied four different deep learning approaches to the question

dataset using Word2vec embedding vectors both skip-gram and CBOW models. We noticed that the use of Word2vec models can efficiently learn the text semantic and syntactic connections between words significantly improved the performance of the classification models. In this research, initially, the Word2vec methods compute the word vectors from vocabulary words and initialized with random vectors. Therefore, by applying a huge amount of text for these algorithms generated by Wikipedia corpus, they will be capable to allow word vectors in the vector space such that their closeness is corresponding to their associated words. By comparative analysis, all deep learning models have revealed superior performance with word2vec models in the tasks of question classification. However, in some of the cases, we observed that the skip gram model performed well as compared to CBOW model.

In our future direction, for further improve the performance we recommend here in one sentence that may motivate us to explore for future work. To improve the accuracy of hybrid feature extraction techniques can be applied for more than one-word embedding methods together for a question classification system. Hopefully, the system will be capable of acquiring the advantages for all embedding techniques when using this hybrid method.

### Funding
This work was supported by the Ministry of Higher Education Malaysia (MOHE) and Universiti Tun Hussein Onn Malaysia for funding this research activity under the Fundamental Research Grant Scheme (FRGS/1/2017/ICT02/UTHM/02/5), vote no. 1641. The funders had no role in study design, data collection and analysis, decision to publish, or preparation of the manuscript.

### Grant Disclosures
The following grant information was disclosed by the authors:
Ministry of Higher Education Malaysia (MOHE).
Fundamental Research Grant Scheme: FRGS/1/2017/ICT02/UTHM/02/5.

### Competing Interests
The authors declare that they have no competing interests.

### Author Contributions
- Muhammad Zulqarnain conceived and designed the experiments, performed the experiments, prepared figures and/or tables, authored or reviewed drafts of the paper, and approved the final draft.
- Ahmed Khalaf Zager Alsaedi conceived and designed the experiments, authored or reviewed drafts of the paper, and approved the final draft.
- Rozaida Ghazali analyzed the data, prepared figures and/or tables, and approved the final draft.

- Muhammad Ghulam Ghouse analyzed the data, performed the computation work, prepared figures and/or tables, and approved the final draft.
- Wareesa Sharif performed the computation work, prepared figures and/or tables, and approved the final draft.
- Noor Aida Husaini performed the computation work, authored or reviewed drafts of the paper, and approved the final draft.

## Data Availability

The code is available at GitHub: https://github.com/zunimalik777/DeepLearning-Turkish-Word2vec-Analysis.

The collected data is available in a Supplemental File.

The metadata is available at Wikimedia: https://dumps.wikimedia.org/trwiki/.

## Supplemental Information

Supplemental information for this article can be found online at http://dx.doi.org/10.7717/peerj-cs.570#supplemental-information.

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
