# Peer review of "A comparative analysis on question classification task based on deep learning approaches"

_PeerJ Computer Science, doi:10.7717/peerj-cs.570_

## Round 0.1 · original submission · Major Revisions

Could you please check the reviewer comments?

All the best

Reviewer 1 ·

Basic reporting

In this manuscript, the authors investigate three-to-five deep learning approaches based on 10-fold cross-validation for question classification tasks in the Turkish language. Kindly suggestions:

Better to move the problem statement (lines 61-65, 78-80) to the right section related to the methodology of this study.

The equations 2-7 with regard to LSTM (long short term memory), and the equations 8-11 about GRU (gated recurrent unit), might contribute limited knowledge.

The setup of the section “METHODOLOGY & RESULTS” might appear a bit unreasonable. That is because the proposed approach and focused novelties should be further improved to facilitate the recognition of other scholars.

The first technique of Wored2vec might not have been well introduced in the context. Further clarification would be grateful.

Experimental design

It's hard to ignore that the section of experiments might not have been well structured. That is to say, the arrangement of this article ought to be further enhanced before the next submission.

Validity of the findings

The novelties of the presented study should be further highlighted in preparing the future version of this manuscript. That might be because innovation is the gold criteria to recognize the most promising study.

Additional comments

The authors have told us a story about their focus on the basis of their deep literature investigation.

Reviewer 2 ·

Basic reporting

The writing needs to be substantially and thoroughly improved and proofread so that it can properly deliver the messages of this work to the reviewers and the readers. There are scattered grammar issues and typos across this manuscript in its current form.
To list a few:
Line 20: There are... have been ... -> ... that have been ...
Line 44: ... and how to ... -> ... (becomes an open question.)
Line 68: ... ... to the credible classify of the answers -> ... credible classification of ...
Line 71: After categories...
Line 96: Most articles focus on English language has ... -> ... focused on ...
Line 123: stag -> stage


The related work part is kind of out-dated, for which the authors may refer to more recent works.

Experimental design

The descriptions of the experimental settings are missing, including the data preprocessing steps, and the training policies and hyper-parameter settings of the deep learning models.
The supplementary code files can not be open properly. You may use UTF-8 for the file encoding.
The publish and organization of a Turkish dataset is definitely meaningful and positive.

Validity of the findings

When saying ``there are several unique features in Turkish languages that make NLP challenging'', please explain it with more details as it's the main motivation of this work.

·

Basic reporting

The whole manuscript should be revised because the quality of written language is very poor making it difficult to understand. It has many grammatical issues and repeated nonsensical text. Even the first sentence of the abstract is an example of this repetition: "Question classification is one of the important tasks for automatic question classification in natural language processing (NLP).". Throughout the text there are too many examples to list. Some technical terms used are also not very clear, for example, the authors say that Turkish is "strongly inflective language" but they also say that it is agglutinative, so this issue should be clarified for the readers.

I would also advise to dismiss most of the text about Word2vec and deep learning models since these are established and widely used algorithms. Instead, more examples about the problem at hand (Turkish question classification) should be given.

The manuscript also has an organization issue: "DEEP LEARNING APPROACHES" starts with an overview of deep learning models but then starts describing the proposed model. Then on "METHODOLOGY & RESULTS", the authors provide a full explanation of the word2vec algorithm, as well as experimental details (but no results since the next section is called "Experimental Results").

Experimental design

The experimental design seems to be an application of existing architectures to turkish question classification. I was unable to understand what adaptations were made to work with Turkish questions. Furthermore very few details are given about the dataset. A dataset was created by translating an english language dataset, but it was not clear how and by whom it was translated and if this dataset is or will be made publicly available. As neither the code or data used to train the models is provided, it would be difficult to replicate the results.

Validity of the findings

The authors provide a comparison between all the architectures considered and different embeddings. They claim that in comparison to english questions, they have lower results due to "an absence of effectiveness lemmatization methods for Turkish Language compared to English language.". However this is the first that they mention lemmatization so I don't know why and how this would impact the results.

Additional comments

This manuscript presents a deep learning approach to question classification applied to turkish questions. The authors compare several deep learning architectures and they mention the challenge that is doing this type of task in Turkish texts. They provide a strong related work section on this task, and describe the models used in detail, however due to language issues most of the manuscript is difficult to understand.

---

## Round 0.2 · Major Revisions

Dear Authors,

Thanks for your efforts to respond to the reviewers' comments, but there are still some issues that still need some improvements. Could you please check the reviewers' feedback?

Reviewer 1 ·

Basic reporting

The quality of this manuscript has apparently improved. Kindly suggest revising the current title of this manuscript as "A comparative analysis on question classification task based on deep learning approaches".

Experimental design

I have no more questions.

Validity of the findings

The current work supports their aims.

Additional comments

Thanks for carefully addressing my concerns.

Reviewer 2 ·

Basic reporting

English writing has been improved, but many issues still exist.

Just to name a few randomly:
Line 267: GRU store[s] and filter[s]
Line 326: to extrcut [extract]
Line 341 we proposed [a] deep learning hybrid architecture

I'd suggest proofreading and revising the manuscript seriously.

Experimental design

It has been moderately improved.

Validity of the findings

It has been moderately improved.

·

Basic reporting

Although there was an overall improvement there are still some language issues with the document. examples L97 covert, L141 to understand to apply, L212 - we evaluated to explain L284 - repeated text.

The inflected/agglutinative issue was also not clarified, there is no explanation of these terms that are used in the text (although inflected now appears only in the abstract). A simple definition would be enough.

The structure was slightly improved although there is still are still explanations of established techniques (peephole LSTM are not new and should be cited) in the proposed methodology section. This could mislead the reader to think the LSTM and GRU variations in this section are new. This article should be cited when refering to peephole LSTMs: "Gers, Felix A., Nicol N. Schraudolph, and Jürgen Schmidhuber. "Learning precise timing with LSTM recurrent networks." Journal of machine learning research 3.Aug (2002): 115-143."

Experimental design

The Turkish dataset was still not properly explained, namely how it was adapted from an English dataset. The authors provide the repository with the code used, although I also could not find the questions corpus there, only a link to the Turkish wikipedia dump.

Validity of the findings

The lines that the authors provide in their response to not match the PDF I have, at least lines 639-648 on the track changes file are something else. Anyway it would be nice if there was a better explanation of the difficulty in processing the Turkish language, such as lemmatization, for those that are unfamiliar with this language. It was interesting to know that it does not have grammatical gender or noun classes but then the techniques used do not seem to differ from the ones used for English.

---

## Round 0.3 · accepted · Accept

Thanks for your response, it is my pleasure to inform you that the amended paper is accepted, thanks for your efforts.

Reviewer 1 ·

Basic reporting

The form of the manuscript looks good at present.

Experimental design

The experimental design of this presented study is credible.

Validity of the findings

The scientific contribution of this paper has been well demonstrated.

Additional comments

Congratulations!

Reviewer 2 ·

Basic reporting

The manuscript has been moderately revised according to the previous reviews. I have no other comments.

Experimental design

I have no other comments.

Validity of the findings

I have no other comments.

Additional comments

I have no other comments.